# Ruminal Methanogenic Responses to the Thiamine Supplementation in High-Concentrate Diets

**DOI:** 10.3390/ani10060935

**Published:** 2020-05-28

**Authors:** Fuguang Xue, Yue Wang, Yiguang Zhao, Xuemei Nan, Dengke Hua, Fuyu Sun, Liang Yang, Linshu Jiang, Benhai Xiong

**Affiliations:** 1State Key Laboratory of Animal Nutrition, Institute of Animal Science, Chinese Academy of Agricultural Sciences, Beijing 100193, China; xuefuguang123@163.com (F.X.); wangyue9313@163.com (Y.W.); zhaoyiguang@caas.cn (Y.Z.); xuemeinan@126.com (X.N.); dengke_h@163.com (D.H.); sfycaas@163.com (F.S.); yangliang@caas.cn (L.Y.); 2Jiangxi Province Key Laboratory of Animal Nutrition/Engineering Research Center of Feed Development, Jiangxi Agricultural University, Nanchang 330045, China; 3Beijing Key Laboratory for Dairy Cow Nutrition, Beijing University of Agriculture, Beijing 102206, China

**Keywords:** high-concentrate diet, metagenomic sequencing, methane, ruminal methanogenesis, thiamine

## Abstract

**Simple Summary:**

Optimal methane (CH_4_) mitigation strategy should decrease carbon footprint and meanwhile avoid causing hazard residues and productive reduction. Thiamine is a critical coenzyme in carbohydrates, and metabolism has been confirmed to effectively attenuate subacute ruminal acidosis (SARA). Whether thiamine supplementation impacts ruminal methanogenesis is still unclear. In the present study, thiamine supplementation in high-concentrate diets (HC) was conducted to investigate the impacts on ruminal methanogenesis process. Results indicated that thiamine supplementation in HC could efficiently reduce CH_4_ emissions compared with high-forage diets; meanwhile, it did not cause ruminal metabolic disorders compared with HC treatment. The effective decrease in methane production and changes of methanogens after thiamine supplementation in HC compared with high-forage diets contributed a better understanding of thiamine on ruminal carbohydrate metabolism. This finding here provided a brand-new perspective on mitigating methane emission in dairy cattle production, and an optimal strategy to decrease the carbon footprint without compromising the livestock production efficiency.

**Abstract:**

Background: Thiamine supplementation in high-concentrate diets (HC) was confirmed to attenuate ruminal subacute acidosis through promoting carbohydrate metabolism, however, whether thiamine supplementation in HC impacts methane metabolism is still unclear. Therefore, in the present study, thiamine was supplemented in the high-concentrate diets to investigate its effects on ruminal methanogens and methanogenesis process. Methods: an in vitro fermentation experiment which included three treatments: control diet (CON, concentrate/forage = 4:6; DM basis), high-concentrate diet (HC, concentrate/forage = 6:4; DM basis) and high-concentrate diet supplemented with thiamine (HCT, concentrate/forage = 6:4, DM basis; thiamine supplementation content = 180 mg/kg DM) was conducted. Each treatment concluded with four repeats, with three bottles in each repeat. The in vitro fermentation was sustained for 48h each time and repeated three times. At the end of fermentation, fermentable parameters, ruminal bacteria and methanogens community were measured. Results: HC significantly decreased ruminal pH, thiamine and acetate content, while significantly increasing propionate content compared with CON (*p* < 0.05). Conversely, thiamine supplementation significantly increased ruminal pH, acetate while significantly decreasing propionate content compared with HC treatment (*p* < 0.05). No significant difference of ruminal methanogens abundances among three treatments was observed. Thiamine supplementation significantly decreased methane production compared with CON, while no significant change was found in HCT compared with HC. Conclusion: thiamine supplementation in the high-concentrate diet (HC) could efficiently reduce CH_4_ emissions compared with high-forage diets while without causing ruminal metabolic disorders compared with HC treatment. This study demonstrated that supplementation of proper thiamine in concentrate diets could be an effective nutritional strategy to decrease CH_4_ production in dairy cows.

## 1. Introduction

Enteric methane (CH_4_) emissions from ruminants were estimated to account for 17–30% of anthropogenic CH_4_ [1], and thus series of CH_4_ mitigation strategies were raised in ruminants production systems to prevent the global warming phenomenon such as monensin [2] and 3-nitrooxypropanol (3NOP) [3]. However, potential negative influences of residual nitro-compounds on rumen metabolism were still in dispute. Therefore, to investigate an optimal methane mitigation strategy, which efficiently decreases the carbon footprint while without negative influences on livestock production efficiency, is of critical demand.

In ruminal conditions, methanogens represent less than 1% of the total microbial population and synergistically interact with bacteria and protozoa in the carbohydrate metabolism process [4]. Methanogens generally produce CH_4_ through the H_2_/CO_2_ reduction pathway, whereas some special methanogens such as *Methanosarcinales* utilize methyl-containing compounds (i.e., methanol and methylamine) for methanogenesis [5]. Although researchers have elucidated the processes of CH_4_ production in rumen [6,7], the complicated bioactivities of methanogens make it difficult to find effective strategies in restricting methanogenesis.

High-concentrate diet (HC) has been well documented to effectively inhibit methanogenesis process by promoting propionate production to mitigate CH_4_ emission from ruminants [5]. However, the continuous feeding of HC changed ruminal microbiota structure and influenced ruminal fermentation characteristics, caused metabolic disorders and even ruminal acidosis [8]. Interestingly, our previous studies discovered that supplementation of an appropriate amount of thiamine in high-concentrate diet alleviated rumen acidosis through increasing abundances of ruminal bacteria and fungi, promoting acetate content while inhibiting ruminal propionate and biogenic amines [9]. As the cofactor of carbohydrate metabolism, thiamine supplementation significantly regulated the ruminal carbohydrate metabolism, however, effects of thiamine supplementation on ruminal methanogenesis were still unclear. We hypothesized that CH_4_ production after thiamine supplementation in HC would not increase because of the decreases in biogenic amines, which are comprised of many methyl-containing compounds.

With the fast development of new molecular techniques, knowledge of rumen microbiology has become proficient in recent years [10,11]. Most methanogens are assigned into three groups at the genus level, which are *Methanobrevibacter*, *Methanomicrobium* and a large group of uncultured ruminal archaea named as rumen cluster C (RCC) [12]. Currently, seven complete genomes have been sequenced for different species of *Methanobacterium* and *Methanobrevibacter* in bovine rumen [6], however, little information is available for the non-cultured methanogens [13]. Moreover, genes that regulate methanogenesis have not been clearly identified. Therefore, a whole metagenome functional analysis is needed to investigate the potential mechanism of methanogenesis. Metagenomic sequencing technology could provide more accurate information on systematic study of the whole rumen microbial community [14] and has been widely applied in studying the phylogeny and functional pathways of rumen microbiome [15]. Therefore, metagenomic sequencing analysis was used in the present study to investigate the effects of thiamine supplementation on methanogens and the relative abundances of genes that regulate CH_4_ metabolism.

## 2. Materials and Methods

Animals and trial procedures used in the present study were in accordance with the recommendations of the academy’s guidelines for animal research, and approved by the Animal Ethics Committee of the Chinese Academy of Agricultural Sciences (Beijing, China); the approval code is IAS 2019-55.

### 2.1. Substrates and Experimental Design

Rumen fluid were obtained from four rumen-cannulated Chinese Holstein Dairy cows which were reared in the dairy cattle farm of the Institute of Animal Science, Chinese Academy of Agricultural Sciences. All cows were fed in the same stall and provided the same diets during trial period, and substrates with detailed ingredients and chemical composition are shown in supplementary Appendix A. The in vitro measurement was conducted in a 75 mL fermentation system for 48h with 0.5 g substrate and 50 mL medium (pH = 6.86) as reported by Wallace R. J. (2001) [16] and 24 mL rumen fluid. Substrates were provided in accordance with our previous in vivo study [17], which included a control diet (CON; concentrate/forage = 4:6, CP = 18.16%, NDF = 36.18%, starch = 20%, DM basis), a high-concentrate diet (HC; concentrate/forage = 6:4, CP = 18.10%, NDF = 27.61%, starch = 30.82%, DM basis) and high-concentrate diet supplemented with 180 mg thiamine/kg DM (HCT, concentrate/forage = 6:4, CP = 18.10%, NDF = 27.61%, starch = 30.82%, DM basis), respectively. Each treatment concluded with 4 repeats, with 3 bottles in each repeat. The in vitro fermentation was repeated three times and the substrates used here were the same as the in vivo feeding diets; the details are shown in Appendix A.

Thiamine supplement was first solved in water to achieve the concentration of 90 μg/mL, and then 1 mL of thiamine solution was added into each fermentation bottle to make sure that the ratio of thiamine and substrate was in accordance with our previous in vivo research (180 mg/kg DM). The other two treatments were added with 1 mL of pure water in their fermentation systems as a control. The gas production data were recorded after 48 h of each fermentation. 

### 2.2. Fermentation Parameters Measurement

Gas samples from each bottle were collected using 10 mL syringes at the end of incubation and gas production of each bottle was measured with a pressure transducer. The CH_4_ concentration was measured by a gas chromatograph (GC-2010, Shimadzu, Kyoto, Japan) [18]. Rumen fluid of each repeat was divided into two portions after incubation finishes. One was conducted to analyze ruminal pH, thiamine content, rumen volatile fatty acids (VFAs), and ammonia-N (NH3-N). The other portion was frozen in the liquid nitrogen immediately, and then, stored at −80 °C for DNA extraction. Portable type pH meter (Testo 205, Testo AG, Lenzkirch, Germany) was applied for the measurement of ruminal pH immediately after the rumen fluid sample was collected. High performance liquid chromatography (HPLC, Agilent 1260 Infinity II Prime, Agilent, San Jose, CA, USA) was conducted to measure thiamine content according to the Analytical Methods Committee (2000). Individual and total VFAs (TVFA) in the aliquots were measured using a gas chromatograph (GC-2010, Shimadzu, Kyoto, Japan). Ruminal fluid was firstly acidified with 250 μL metaphosphoric acid, one-tenth volume of rumen fluid, centrifuged with 3000 rpm, and injected directly on to a column which containing a porous aromatic polymer (Chromosorb 101). The sample was maintained at 200 °C in a gas chromatograph and fitted with a flame ionization detector [19]. The concentration of NH3-N was determined by the indophenol method [20]. A standard curve of NH3-N content was first created by using standard NH_4_Cl. The absorbance was adjusted through a pure water without NH3-N content at the 700 nm wavelength. Finally, the absorbance value was measured through the UV-2600 ultraviolet spectrophotometer (Tianmei Ltd., Shanghai, China).

### 2.3. Metagenomic Sequencing Process

Metagenomic sequencing methods were detailed described in our previous study [21]. To simply state here, DNA was extracted using the QIAamp DNA Stool Mini Kit (Qiagen, Hilden, Germany), and TBS-380 (P/N,3800-003, Turner Biosystem, Mary Ave Sunnyvale, CA, USA) and NanoDrop2000 (Thermo Fisher Scientific, Waltham, MA, USA) were applied for the measurement of the concentration and purity of DNA, respectively. DNA quality was examined by a 1% agarose gel electrophoresis system, while a DNA library was constructed using the TruSeqTM DNA Sample Prep Kit (Illumina, San Diego, CA, USA). The Illumina HiSeq 4000 platform (Illumina Inc., San Diego, CA, USA) combined with a HiSeq 3000/4000 PE Cluster Kit (www.illumina.com) was used for the paired-end sequencing process.

SeqPrep (https://github.com/jstjohn/SeqPrep) [22] was applied for the quality controlling of primary reads while SOAPdenovo (http://soap.genomics.org.cn, Version 1.06), which worked based on De Brujin graph construction, was used for the filtering of reads [23]. Scaffolds with a length less than 500 bp were removed and the retained were applied for the extraction of contigs without gaps. Contigs were used for further gene prediction and annotation. Open reading frames (ORFs) from each sample were predicted using MetaGene (http://metagene.cb.k.u-tokyo.ac.jp/) [24]. The predicted ORFs with length being or over 100 bp were translated to amino acid sequences using the NCBI translation table (http://www.ncbi.nlm.nih.gov/Taxonomy/taxonomyhome.html/index.cgi?chapter=tgencodes#SG1). All sequences from gene sets with a 95% sequence identity (90% coverage) were clustered as the non-redundant gene catalog by the CD-HIT (http://www.bioinformatics.org/cd-hit/). Reads after quality control were mapped to the representative genes, with 95% identity using a SOAP aligner (http://soap.genomics.org.cn/), and gene abundance in each sample was evaluated [25]. Taxonomic annotations through alignment of non-redundant gene catalogs against the NCBI NR database with the selection cutoff of 1 × 10^−5^ and score >60 were conducted by DIAMOND [26]. Functional investigation of pathway annotation was conducted by GhostKOALA (https://www.kegg.jp/ghostkoala/) [27] against the Kyoto Encyclopedia of Genes and Genomes database (http://www.genome.jp/kegg/) with a cutoff of 1 × 10^−5^. 

### 2.4. Statistical Analysis 

Ruminal pH, thiamine content, ruminal fermentation variables, total gas and CH4 production were firstly conducted a normal distribution test using SAS procedure “proc univariate data = test normal” and subsequently, one-way ANOVA S-N-K test was applied to investigate the differences among the three treatments. Significance would be considered when *p* < 0.05, while a tendency was considered when 0.05 ≤ *p* < 0.10. Correlation analysis between methanogens at the level of phyla and 30 most abundant species and ruminal fermentable parameters, thiamine content were assessed firstly using the PROC CORR procedure of SAS 9.2 (SAS Institute, Inc., Cary, NC, USA) to correlate the correlation matrix, and then, the Spearman correlation matrix was visualized in a heatmap format using R (Version 3.3.1, R Core Team, Vienna, Austria) “pheatmap package”. The absolute value of correlation coefficients (|r| > 0.55 and *p* < 0.05) was considered to be correlated between the abundance of methanogens communities and ruminal variables. Principal coordinate analysis (PCoA) for different rumen methanogens were conducted using R (version 3.3.1) “vegan package”. 

## 3. Results

### 3.1. Rumen Fermentation Parameters and Methane Production

As shown in Table 1, ruminal pH value, thiamine and acetate content were significantly decreased while propionate and ammonia-N were significant increased after HC treatment. However, these changes were significantly inversed by thiamine supplementation (*p* < 0.05). Methane production significantly decreased in HC treatment (*p* < 0.05) compared with CON, while no significant change was detected after thiamine supplementation (*p* > 0.05). 

### 3.2. Taxonomy Results of Ruminal Bacteria and Methanongens

The relative abundances of ruminal bacteria ranged from 75–80% of total microbiome, in which, we identified 32 phyla and numbers of candidate phyla. Bacteroidetes were the most abundant phylum accounting for 45–60% of the whole bacteria. Firmicutes were the second most abundant phylum accounting for 15–25% of the total bacteria. At genus level, *Prevotella* was the most abundant genus accounting for 25–35% of the total bacteria. Besides, 4 classes, 30 genus and more than 100 species of methanogens are identified in the present study. The relative abundances of ruminal methanogens ranged from 0.37% to 0.47%, and all methanogens were classified in the phylum of *Euryarchaeota. Methanobrevibacter* was the most abundant genus and accounted for 50%~55% of the total methanogens. *Thermoplasmatales-affiliated Lineage C*(TALC) which had not been characterized yet, accounted for 31% of the total methanogens. *Methanococci*, *Methanomicrobia* and *Methanopyri* together accounted for about 15% of the total methanogens. These results are shown in Appendix A. 

### 3.3. Effect of Thiamine Supplementation in High-Concentrate Diets on Ruminal Methanogenesis 

High-concentrate diet feeding significantly suppressed the proliferation of *Bacteroidetes* and *Fibrobacteres*, while significantly increasing the abundances of *Firmicutes* and *Proteobacteria*. Thiamine supplementation significantly inversed all these changes. These results are shown in Table 2.

For methanogens, principal coordinates analysis (PCoA) was conducted to compare methanogens profile in the three treatments, and this result is shown in Figure 1. PCoA axes 1 and 2 accounted for 66.86% and 19.13% of the total variation, respectively. Based on the PCoA results, methanogens from the three treatments are clearly separated; methanogens in HC treatment were significantly separated from those in the CON and HCT treatments by PCo1 and methanogens in the CON treatment were significantly separated from those in HCT treatment by PCo2.

Differential analysis of the genera and species of methanogens were then conducted to investigate which community was influenced by HC and thiamine supplementation. At the genera level, as shown in Table 3, total relative abundance of methanogens was not affected by HC and thiamine supplementation treatments. HC treatment significantly reduced the relative abundances of *Methanobrevibacter* and *Methanobacterium*, yet, significantly increased the relative abundance of TALC and *Methanoculleus* compared with CON (*p* < 0.05). Thiamine supplementation significantly increased the relative abundances of *Methanobrevibacter* and *Methanobacterium* and decreased TALC and *Methanoculleus* compared with HC (*p* < 0.05), however, the relative abundance of *Methanobrevibacter* was significantly lower than that of in CON (*p* < 0.05). Specifically, *Methanocaldococcus* was significantly increased in HCT compared with both CON and HC (*p* < 0.05).

Spearman correlation analysis was conducted between ruminal VFAs, DMI, thiamine content and ruminal methanogens, respectively. As shown in Figure 2, methanogens were classified into two clusters based on their relationships with the acetate and propionate concentrations. *Methanocaldococcus*, *Methanobrevibacter*, *Methanobacterium*, *Methanosarcina*, *Methanosaeta*, and *Methanococcus* were positively correlated with acetate while *Methanocaldococcus*, *Methanobrevibacter* and *Methanobacterium* had negative correlations with propionate and isovalerate. TALC and *Methanoculleus* showed inverse correlations with propionate and acetate. Thiamine had positive correlations with *Methanobrevibacter* and *Methanobacterium*, yet, tended to be negatively correlated to TALC and *Methanoculleus*.

### 3.4. Functional Analysis of Thiamine Supplementation on Ruminal CH4 Metabolism 

The CH_4_ metabolism pathway was listed separately for further analysis in order to investigate the related genes that affected by HC and thiamine supplementation. The results (shown in Figure 3; Figure 4) indicated that the abundances of genes encoding formate dehydrogenase and gamma-F420-2 were significantly increased while gene encoding heterodisulfide reductase, heterodisulfide reductase, coenzyme F420 hydrogenase subunit alpha, and glycine hydroxymethyltransferase were significantly decreased in the HC treatment compared with CON. The abundances of genes that encoded heterodisulfide reductase and 2-phosphosulfolactate phosphatase were significantly increased, whereas those genes encoding formate dehydrogenase, coenzyme M methyltransferase (MtaA), trimethylamine corrinoid protein (MttC) and gamma-F420-2 were significantly decreased after thiamine supplementation.

## 4. Discussion

### 4.1. Effects of Thiamine Supplementation in High-Concentrate Diets on Ruminal Methanogens

In the present study, relative numbers of ruminal methanogens ranged from 0.37% to 0.47%. This is in line with W Adg and AV Klieve [4], in which methanogens were reported to accounted for less than 1% of the total microbial population. *Methanobrevibacter* dominated the most abundant methanogens in the current study, which agreed with the finding of EE King [24]. The optimum temperature and pH for the growth of *Methanobrevibacter* ranged from 37 to 40 °C and 6.5 to 7.0, respectively [28], which were just the normal body temperature and ruminal pH of cows. The significantly dropped ruminal pH in HC treatment limited the activity of *Methanobrevibacter*, which might contribute to reducing the CH_4_ production. An uncultured genus belonged to *Thermoplasmata* was identified as the second most abundant methanogen and significantly increased in HC treatment compared with the other two treatments. This special genus was reported to be referred as the *Thermoplasmatales-affiliated Lineage C* (TALC) and hold a strong tolerance ability of extreme environment [13]. The endurance capacity ensures its activity in the low pH environment and resulted in its increased relative abundance in HC treatment.

Thiamine supplementation did not significantly increase the relative abundance of methanogens. However, it raised the ruminal pH [29], which might promote the abundances of *Methanobrevibacter* and enriched the relative abundances of the rumen fungi which was reported to have synergistic effect with methanogens [30]. Besides, acetate was also reported the main substrate in rumen methanogenesis, and acetate to propionate ratio was demonstrated to positively correlated with methane emission [27]. The acetate to propionate ratio in the present study significantly increased after thiamine supplementation, which indicated a potential increase in methane emission might be detected. Conversely, thiamine significantly decreased the abundances of *Methanosarcinales* which were reported to utilize methanol and methylamines as substrates [31] because of the reduction in ruminal biogenic amines after thiamine supplementation [9], and the ruminal TALC. This contrary effect may attribute to that thiamine supplementation did not significantly increase the relative abundance of methanogens in the present study.

### 4.2. Effects of Thiamine Supplementation on CH_4_ Metabolism and CH_4_ Production

Although thiamine supplementation increased ruminal acetate content, promoted ribulose-P pathway and improved activities of coenzyme B and coenzyme M, it did not significantly increase the CH_4_ production. In ruminal conditions, as shown in the previous study, methane could be produced through CO_2_, acetate, methylamine etc. [32]. Thiamine supplementation significantly increased acetate, yet, reduced ruminal biogenic amines content, which might suppress CH_4_ production. In addition, based on the KEGG results, the abundance of genes that encoded formate dehydrogenase and gamma-F420-2 were significantly decreased after thiamine supplementation which indicated that the formate utilization and activity of coenzyme F420 were inhibited. In addition, genes that encoded MttC which mainly participate in the transportation of methyl groups from methylamine were decreased after thiamine supplementation. As a common methyl donor, methylamine also contributes to the CH_4_ production. Thiamine has been proved to reduce the ruminal biogenic amines content [9], which might lead to the methylamine content decreased, and therefore, induced the decreased abundance of MttC genes. This may also partly explain that CH_4_ production was not significantly increased after thiamine supplementation.

### 4.3. Comparison of Thiamine Supplementation in HC with Other CH_4_ Mitigation Strategies

Nutritional strategies have been developed to mitigate CH4 emissions from ruminants such as supplementation of tannins and saponins [33], lipid addition [34], and 3-nitrooxypropanol (NOP) [35] as feed additives in diets. However, substantial mitigation of CH_4_ emissions in husbandry of these methods are rarely proved to be practical and cost-effective. Despite acting as electron sinks and directly inhibited the methanogens, dietary nitrate supplementation may increase the risk of nitrite toxicity [36]. NOP was proved to competitively inhibit the methyl-coenzyme M reductase, which is the last step in the formation of CH_4_, resulted in effectively reducing CH_4_ production and inhibiting the growth of methanogens [3]. However, “where did the nitro go” is still a risk for the health of dairy cattle. The CH_4_ mitigation options with both nutritional and environmental advantages would be better for the long-term sustainable mitigation. In the present study, thiamine supplementation in HC diets may also provide a novel choice for the benefits of both animal performance and environment footprint.

## 5. Conclusions

In summary, high-concentrate diet feeding was conclusively proved to reduce methane production through increasing ruminal propionate content and decreasing acetate to change ruminal fermentable patterns. However, ruminal acidosis occurred frequently by HC feeding procedure. The combination of thiamine and HC diets attenuated SARA, which is always induced by high-concentrate diets through regulating ruminal fermentable patterns, while reducing CH_4_ emission when compared with high-forage diet through regulating ruminal bacteria and methanogens. The findings in this study could therefore contribute to the further understanding of the mechanism of thiamine’s function in dairy cows and may help reduce the economic loss caused by HC induced SARA while decreasing CH_4_ production and the greenhouse effect.

## Figures and Tables

**Figure 1 animals-10-00935-f001:**
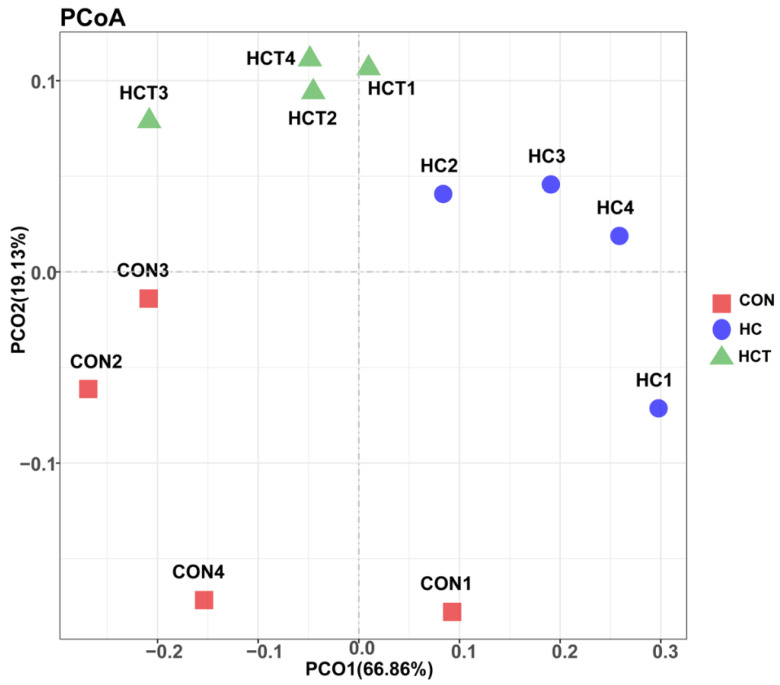
Principal coordinate analysis (PCoA) on ruminal methanogens community structures of CON, HC, and HCT treatments. CON = control diet, concentrate/forage = 4:6, CP = 18.16%, NDF = 36.18%, starch = 20%, DM basis; HC = high-concentrate diet, concentrate/forage = 6:4, CP = 18.10%, NDF = 27.61%, starch = 30.82%, DM basis; HCT = high-concentrate diet supplemented with 180 mg thiamine/kg DMI.

**Figure 2 animals-10-00935-f002:**
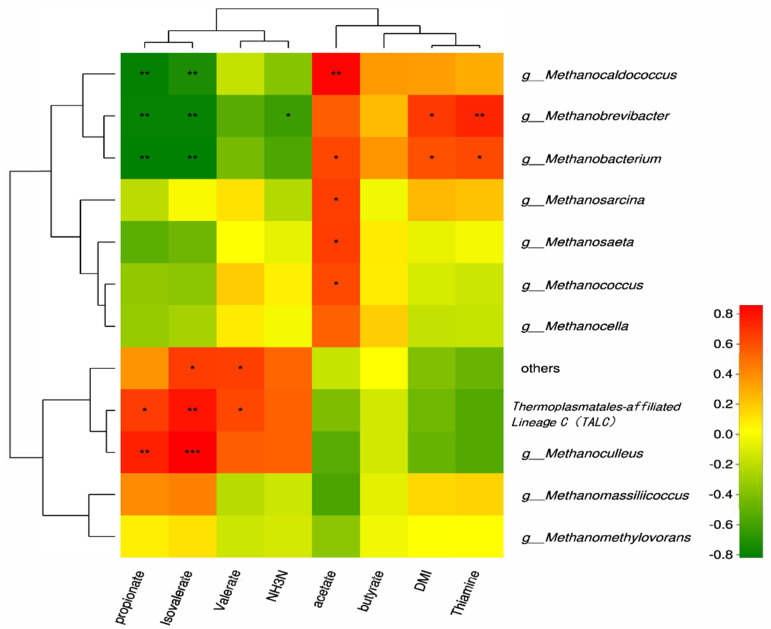
Correlation analyses between relative abundances of methanogen genus, ruminal VFA, thiamine concentrations, and DMI. The red blocks mean positive correlations; the green blocks mean negative correlations. “*” means a significant correlation (|r| > 0.55, *p* < 0.05), “**” means a strong correlation (|r| > 0.75, *p* < 0.05), and “***” means the extremely significant correlation (|r| > 0.75, *p* < 0.01).

**Figure 3 animals-10-00935-f003:**
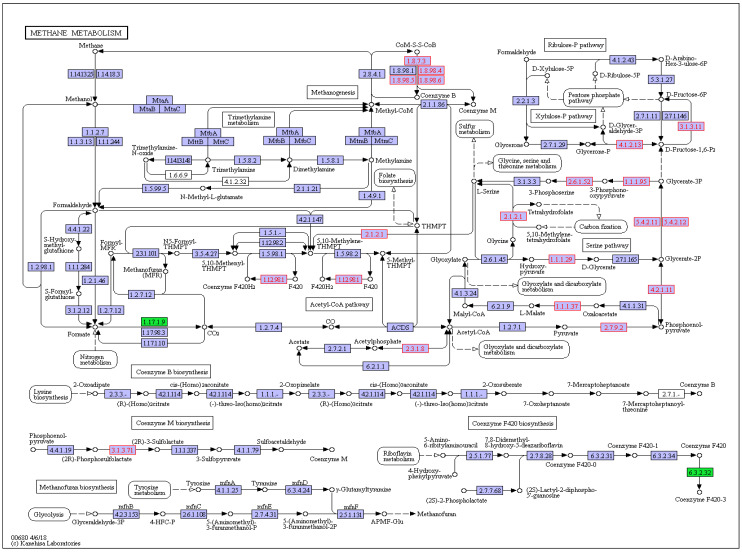
Different genes’ abundances that enriched in methane metabolism of CON vs. HC. CON = control diet, concentrate/forage = 4:6, CP = 18.16%, NDF = 36.18%, starch = 20%, DM basis; HC = high-concentrate diet, concentrate/forage = 6:4, CP = 18.10%, NDF = 27.61%, starch = 30.82%, DM basis. The circles represent intermediate metabolites of methane metabolism and the blocks represent enzymes that participate in the methane metabolism. Red blocks mean the increased gene’s abundance and green blocks mean the decreased gene’s abundance.

**Figure 4 animals-10-00935-f004:**
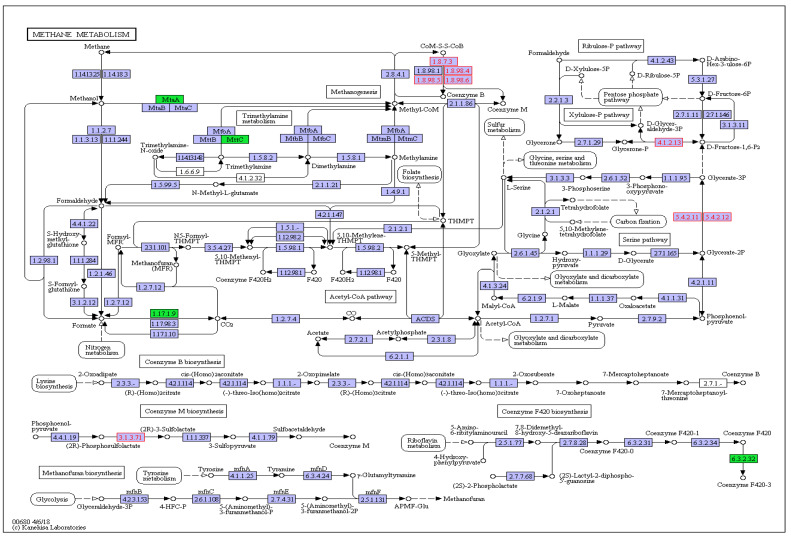
Different genes’ abundances that enriched in methane metabolism of HCT vs. HC. HC = high-concentrate diet, concentrate/forage = 6:4, CP = 18.10%, NDF = 27.61%, starch = 30.82%, DM basis, HCT = high-concentrate diet supplemented with 180 mg thiamine/kg DMI. The circles represent intermediate metabolites of methane metabolism and the blocks represent enzymes that participate in the methane metabolism. Red blocks mean the increased gene’s abundance and green blocks mean the decreased gene’s abundance.

**Table 1 animals-10-00935-t001:** Effects of high-concentrate diet feeding and thiamine supplementation on rumen fermentation parameters and methane production.

Items	Experimental Treatments			SEM	*p*-Value
	CON	HC	HCT		
Ruminal pH	6.35 ^a^	5.88 ^c^	6.12 ^b^	0.094	0.016
Thiamine content (ug/L)	12.16 ^a^	8.51 ^b^	13.53 ^a^	1.933	<0.001
Acetate (mmol/L)	44.24 ^a^	41.62 ^b^	44.07 ^a^	1.273	0.038
Propionate (mmol/L)	12.49 ^b^	13.75 ^a^	11.44 ^c^	0.632	0.027
Propionate/Acetate	0.282 ^b^	0.330 ^a^	0.259 ^b^	0.023	0.043
Butyrate (mmol/L)	10.77	10.35	10.82	0.137	0.356
Ammonia-N (mg/100 mL)	11.49 ^b^	13.96 ^a^	11.27 ^b^	1.711	0.006
Total gas production (mL·g^−1^ DM)	118.4 ^a^	105.2 ^b^	112.3 ^ab^	3.444	0.016
Methane (mL·g^−1^ DM)	6.91 ^a^	5.48 ^b^	5.82 ^b^	0.416	0.001

^a,b,c^ means within a row with different letters differed significantly (*p* < 0.05); SEM, standard error of the mean. CON = control diet; HC = high-concentrate diet); HCT = high-concentrate diet supplemented with thiamine; TVFA = total volatile fatty acid.

**Table 2 animals-10-00935-t002:** Effects of high-concentrate diet feeding and thiamine supplementation on relative abundances of rumen bacteria (%).

Items	Experimental Treatments			SEM	*p*-Value
	CON	HC	HCT		
*Bacteroidetes*	55.78 ^a^	31.51 ^c^	39.91 ^b^	3.274	0.001
*Firmicutes*	10.26 ^b^	15.57 ^a^	8.86 ^b^	0.935	0.001
*Bacteria_noname*	6.34 ^a^	3.32 ^b^	3.89 ^b^	0.459	0.002
*Proteobacteria*	5.06 ^b^	6.83 ^a^	3.25 ^a^	0.769	0.169
*Fibrobacteres*	0.99 ^a^	0.64 ^b^	0.88 ^a^	0.058	0.028
*Spirochaetes*	0.95	1.00	0.72	0.066	0.187
*Actinobacteria*	0.42	0.71	0.60	0.044	0.008
*Verrucomicrobia*	0.32	0.34	0.47	0.031	0.07
*Chlamydiae*	0.21	0.59	0.23	0.059	0.001
*Cyanobacteria*	0.19	0.14	0.54	0.056	0.001
*Planctomycetes*	0.10	0.08	0.16	0.012	0.024
*Tenericutes*	0.08	0.12	0.19	0.017	0.001
*Fusobacteria*	0.07	0.12	0.11	0.011	0.158
others	0.38	0.44	0.49	0.016	0.002

The number a,b,c means within a row data was significantly different (*p* < 0.05); SEM = standard error of the mean. CON = control diet treatment, HC = high-concentrate diet treatment, HCT = high-concentrate diet supplemented with 180 mg thiamine/kg DM treatment.

**Table 3 animals-10-00935-t003:** Effects of high-concentrate diet (HC) feeding and thiamine supplementation (HCT) on relative abundances of methanogens in rumen fluid (%).

Methanogens	Treatments			SEM	*p*-Value
	CON	HC	HCT		
*Methanobrevibacter*	0.1330 ^a^	0.0429 ^c^	0.0996 ^b^	0.0139	0.008
*Thermoplasmatales-affiliated Lineage C(TALC)*	0.0916 ^b^	0.1723 ^a^	0.0970 ^b^	0.0155	0.040
*Methanomassiliicoccus*	0.0376	0.0129	0.0068	0.0067	0.131
*Methanosarcina*	0.0191	0.0177	0.0241	0.0014	0.160
*Methanobacterium*	0.0099 ^a^	0.0032 ^b^	0.0089 ^a^	0.0012	0.036
*Methanoculleus*	0.0068 ^b^	0.0115 ^a^	0.0066 ^b^	0.0010	0.050
*Methanocaldococcus*	0.0031 ^b^	0.0017 ^c^	0.0062 ^a^	0.0006	0.001
*Methanococcus*	0.0015 ^b^	0.0021 ^b^	0.0064 ^a^	0.0007	<0.001
*Methanosaeta*	0.0014 ^b^	0.0016 ^b^	0.0030 ^a^	0.0002	0.006
*Methanocella*	0.0014 ^b^	0.0018 ^b^	0.0063 ^a^	0.0007	<0.001
others	0.0512	0.0740	0.0652	0.0046	0.123
Total	0.3581	0.3429	0.3304	0.1553	0.783

^a,b,c^ means within a row with different letters differed significantly (*p* < 0.05); SEM = standard error of the mean, CON = control diet, concentrate/forage = 4:6, CP = 18.16%, NDF = 36.18%, starch = 20%, DM basis; HC = high-concentrate diet, concentrate/forage = 6:4, CP = 18.10%, NDF = 27.61%, starch = 30.82%, DM basis; HCT = high-concentrate diet supplemented with 180 mg thiamine/kg DMI.

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
