# Peer review of "Ruminal Methanogenic Responses to the Thiamine Supplementation in High-Concentrate Diets"

_animals, 2020, doi:10.3390/ani10060935_

Round 1

Reviewer 1 Report

The conclusions are not clearly supported by the results, especially with the association between thiamine and HC, HC and HCT does not have differences in methane production, but if it reduces propionic, increases acetic, and there is an important modification in the microbial populations.

Author Response

Dear reviewer, Thank you for the time you paid on the present article. Based on the comments, we corrected the conclusion and other sections that was not decribed well. We hope these changes could meet the requirement

Reviewer 2 Report

Dear authors,

I trust the work to be relevant to Animal Science and I believe it warrants consideration for publication. However, I feel that as it is, the manuscript lacks clarity and needs more work in specific points in order to not only clarify the ideas you want to pass but also to improve the overall quality and meet the standards of this Journal.

Below I list some comments that will hopefully assist the authors with improvements on quality of the final draft:

Lines 19 and 20: What do you mean when you say: “Thiamine supplementation in HC could efficiently reduce CH4 emission without causing ruminal metabolic disorders compared with high-forage diets”? Question: Thiamine was added only to HC, right? Or also to the high-forage control diet? I don’t think that is the case. Be careful with your assumptions. Ex: You are referring to acidosis, right? Why would it be a problem in high-forage diets? This paragraph seems a bit confusing to me.

Line 27: What is limited is the “knowledge” on thiamine impacts, right? When you say limited impacts, it means it does not impact much. Please verify.

Line 30: You should give a very brief explanation of what your control diet is. At least say high-forage diet, not just control diet, but preferably describe the ratios of concentrate:forage. For a reader that works with feedlot beef cattle, a HC would have 85-90% concentrate. The details you have in M&M line 97 should be given here at the start. And perhaps you need to add more details in M&M. You say 33.2% starch and I assume that corn is your main ingredient, but you could have other ingredients high in pectin for example. Use the M&M to really explain in details what you did. NOTE: I see you have a supplementary table with feed ingredients. Despite this I would have more details at least of chemical composition such as CP, NDT, NDF.

Lines 35 and 36: OK, the pH and acetate were decreased in comparison to control. Note that here the reader has to assume that your control diet is high in forage. And then you say that thiamine supplementation inversed this. What do you mean? The pH and acetate were increased in the HC in comparison to HCT or to control diet? Please clarify.

Line 37: Methane emissions decreased in HC and HCT in comparison to control. Again, the reader at this point has to assume your control diet has a higher proportion of forage and yes, the methane emissions would be expected to drop with higher allocations of concentrate because of modifications on fermentation patterns. I think here it is important to emphasise that the addition of thiamine did not affect methane emission. Note: that was your main goal, right? To find out the influences of thiamine.

Line 39: Thiamine supplementation could efficiently reduce CH4 emission? Please verify, that was not what you presented in line 38.

Line 50: Residual “nitro-“?

Lines 90-101: The details on description of the in vitro procedure are sound and clear but my previous comment regarding the diets per se still stands.

Line 107: Please check the spelling “parameters”.

Line 117: Check if you can start a sentence with abbreviations (VFA). You may need to write “volatile fatty acids” in full at the beginning of the sentence.

Line 123: Same as above. Check with editorial service what the norm is.

Table 1: I see here that thiamine has influenced on acetate production but had no effects on total gas production. Please consider revising your statement on line 40. Did thiamine increase CHO utilization or it changed the fermentation pattern? Note: The changes in propionate:acetate ratio are a good thing. I would recommend authors to consider the addition of a row in Table 1 with propionate:acetate ratio.

Line 189: OK Now I understand what you meant in the Abstract when you said that thiamine “inversed: those changes. Please consider going back to line 36 to rewrite that paragraph so it is clear for future readers.

Lines 186 to 255: A lot of emphasis was put into the microbiology results. Good job, everything is sound and clear in this session.

Line 256: Discussion: in general, the discussion is well conducted and it is clear that a lot of emphasis has been put into the microbial species in all of the treatment diets. I caution authors to not disregard other important aspects of your work. Example: Line 280, item 4.2. Your subheading says effects of thiamine on CH4 metabolism and CH4 production. I know it is the central part of your work, but as previously mentioned, you looked at “rumen parameters” and you have gathered a considerable amount of information on VFA changes, pH, etc. Of course in the end you want to see the effects on CH4 emission but you can discuss other variables that you collected along the way.

Lines 308 to 316: Note: In your conclusion, what has in fact attenuated methane emission is HC vs high forage and not thiamine. Despite this, it is clear that thiamine can bring benefits by changing the fermentation pattern what could potentially decrease issues arisen from the use of HC diets (ex. Acidosis). I completely agree. I do think that this has to be clarified earlier in the manuscript and not just at the end. Remember that many readers only glance at papers and therefore things need to be clear in the Abstract or even in the Title. Scientific papers do not give much room for suspense.

Author Response

Dear reviewer, Thank you for the time you paid on the present article and thank you for your kindly suggestions. Based on the comments, we corrected the article, make some changes and we hope these changes could meet the requirement. Best regards
